

# 1  A multiplexing system for quantifying oxygen fractionation factors in
# 2  closed chambers

Clémence Paul[1], Clément Piel[2], Joana Sauze[2], Olivier Jossoud[1], Arnaud Dapoigny[1], Daniele Romanini[4],
Frédéric Prié[1], Sébastien Devidal[2], Roxanne Jacob[1], Alexandru Milcu[2,3], Amaëlle Landais[1]
[1] Laboratoire des Sciences du Climat et de l'Environnement, LSCE - IPSL, CEA-CNRS-UVSQ, Université Paris-
Saclay, 91191 Gif-sur-Yvette, France
[2] Ecotron Européen de Montpellier (UAR 3248), Univ Montpellier, Centre National de la Recherche Scientifique
(CNRS), Campus Baillarguet, Montferrier-sur-Lez, France
[3] CEFE, Univ Montpellier, CNRS, EPHE, IRD, Montpellier, France
[4] Laboratoire Interdisciplinaire de Physique, Univ Grenoble Alpes, CNRS/UGA, Saint-Martin-d'Hères, France
Correspondence: Clémence Paul (clemence.paul@lsce.ipsl.fr)
Abstract
The study of isotopic ratios of atmospheric oxygen in fossilized air trapped in ice core bubbles provides
information on variations in the hydrological cycle at low latitudes and productivity in the past.
However, to refine these interpretations, it is necessary to better quantify fractionation of oxygen in
the biological processes such as photosynthesis and respiration. We set up a system of closed biological
chambers in which we studied the evolution of elemental and isotopic composition of $O_2$ due to
biological processes. To easily replicate experiments, we developed a multiplexing system which we
describe here. We compared measurements of elemental and isotopic composition of $O_2$ using two
different measurement techniques: optical spectrometry (Optical-Feedback Cavity- Enhanced
Absorption Spectroscopy, i.e. OF-CEAS technique), which enables higher temporal resolution and
continuous data collection and isotopic ratio mass spectrometry (IRMS) with a flanged air recovery
system, thus validating the data analysis conducted through the OF-CEAS technique. As a first
application, we investigated isotopic discrimination during respiration and photosynthesis. We
conducted a 5-day experiment using maize (*Zea mays* L.) as model species. The [18]O discrimination value
for maize during dark plant respiration was determined as - 17.8 ± 0.9 ‰ by IRMS and - 16.1 ± 1.1 ‰
by optical spectrometer. We also found a value attributed to the isotopic discrimination of terrestrial



photosynthesis equal to + 3.2 ± 2.6 ‰ by IRMS and + 6.7 ± 3.8 ‰ by optical spectrometer. These
findings were consistent with a previous study by Paul et al. (2023).






























1. Introduction

Oxygen, the most abundant chemical element on Earth, is present in all the geological layers, both
internally and externally. In the surface layers of the Earth (atmosphere, biosphere, ocean), it is
produced from water through the well-known biological process of photosynthesis. Consumption of
$O_2$ is mainly due to respiration. These fluxes are responsible for the seasonal variations of dioxygen
concentration in the atmosphere (Keeling and Shertz, 1992) and play a role in the longer-term
evolution of $O_2$ (Stolper et al., 2016). Oxygen consists of three stable isotopes: $^{16}O$, $^{17}O$ and $^{18}O$. By
measuring the ratios of these isotopes, we can document the physicochemical and biological processes
involved in the oxygen cycle. We use the $\delta^{18}O$ notation to express the isotopic signal of oxygen
compared to a reference isotopic ratio (Eq. 1):

$$\delta^{18}O_{calibrated} = \left[ \frac{\left(\frac{^{18}O}{^{16}O}\right)_{sample}}{\left(\frac{^{18}O}{^{16}O}\right)_{standard}} - 1 \right] \times 1000 \qquad (1)$$


Oxygen isotopes do not have the same thermodynamic properties. Thus, during phase changes,
fractionation occurs which is measured by the fractionation factor $\alpha$ (Eq. 2):

$$^{18}\alpha = \frac{^{18}R_{product}}{^{18}R_{substrate}} \qquad (2)$$


where $^{18}R$ is the ratio of the concentration $^{18}R = \frac{n(^{18}O)}{n(^{16}O)}$ and $n$ the number of moles of $O_2$ containing
$^{18}O$ or $^{16}O$.
The isotopic discrimination is related to the isotopic fractionation factor through:

$$^{18}\varepsilon = {}^{18}\alpha - 1 \qquad (3)$$


The isotopic composition of dioxygen in the atmosphere, $\delta^{18}O$ of $O_2$ in air, is often noted $\delta^{18}O_{atm}$. This
signal, measured in the air bubbles in ice cores, can be related to the past variations of the hydrological
cycle of water in the low latitudes, the relative proportion of oceanic vs terrestrial productivity as well



as to the biosphere productivity (Bender et al., 1994; Luz et al., 1999; Severinghaus et al., 2009;
Brandon et al., 2020; Yang et al., 2022). The reconstruction of the relative proportion of oceanic vs
terrestrial productivity can be done using $\delta^{18}O_{atm}$ only as long as the fractionation coefficients of $^{18}O$ /
$^{16}O$ associated with biological processes are known. The second application (biosphere productivity
reconstruction) relies on the observation that biological productivity processes (respiration and
photosynthesis) fractionate oxygen in a mass dependent manner (i.e. there is a consistent relationship
between changes in $\delta^{17}O$ and $\delta^{18}O$, approximately equal to 0.5), while dioxygen originating from
exchanges with the stratosphere has an isotopic composition affected by mass independent
fractionation (hence a relationship between changes in $\delta^{17}O$ and $\delta^{18}O$ significantly different from 0.5
i.e. between 1 and 2). The relative proportion of biosphere productivity vs stratospheric exchange
fluxes sets the value of the relationship between $\delta^{17}O$ vs $\delta^{18}O$ in the troposphere, which is often
described as $\Delta^{17}O = \ln(1 + \delta^{17}O) - 0.516 \times \ln(1 + \delta^{18}O)$ (Luz et al., 1999). In parallel, the same
parameter $\Delta^{17}O$ measured in the air dissolved in the ocean permits to constrain the gross biosphere
productivity when combined with the concentration of $O_2$ measured as the ratio $O_2/Ar$ (Luz et al.,

104  2000).

Despite our system can in theory enable determination of the triple isotopic composition of $O_2$
(through IRMS, Isotopic Ratio Mass Spectrometry), we will focus on $\delta^{18}O$ of $O_2$ in the present study.
We thus concentrate on fractionation coefficients needed to interpret $\delta^{18}O_{atm}$ records only.
Several studies conducted over the years at the cell level (Guy et al., 1993; Angert et al., 2001; Helman
et al., 2005; Eisenstadt et al., 2010; Stolper et al., 2018) have revealed variations in oxygen
fractionation among different biological species and methods employed. Guy et al. (1993) conducted
investigations on spinach thylakoids, cyanobacteria (*Anacystis nidulans*) and diatoms (*Phaeodactylum*
*tricornutum*), and estimated a respiratory discrimination of oxygen by about 21 ‰. Kroopnick and
Craig (1972) measured this effect on plankton incubated in natural seawater and obtained a similar
value. Luz and Barkan (2002) found a respiratory fractionation of 21.6 ‰ on incubation experiments
with natural plankton in Lake Kinneret. Finally, the global average oceanic respiratory fractionation
value given by Luz and Barkan (2011) is 19.7 ‰ on samples from the Celtic Sea, Southern Ocean, North
Atlantic and Red Sea. For terrestrial respiration, using a compilation of values from previous
experiments, Bender et al. (1994) gave a global respiratory fractionation value of 18 ‰. Angert et al.
(2001) focused on soil samples and gave a soil respiratory fractionation (roots and micro-organisms)
of around 12 ‰. This lower value is the result of the role of roots in limiting oxygen diffusion in the
consumption site.



Guy et al. (1993) showed that photosynthesis does not fractionate oxygen between the water
consumed and the dioxygen produced by the organism. However, Eisenstadt et al. (2010) found later
a discrimination up to 6 ‰ for oceanic photosynthesis on a study on oceanic phytoplankton, whereas
Paul et al. (2023) found a discrimination of 3.7 ± 1.3 ‰ for terrestrial photosynthesis with an
experiment performed at the scale of a terrarium with *Festuca arundinacea.*
The variety of values found for the different studies can be attributed to the different set-up used,
different environment or different species. To determine robust values of fractionation coefficients, it
is necessary to proceed in a systematic way and use the same set-up for a large variety of plants and
environments.
In this study, we present an automated setup which can be used to perform numerous systematic
studies of the fractionation factor of oxygen during biological processes. Similar to the study of Paul et
al. (2023), we used closed growth chambers to quantify oxygen fractionation factors associated with
respiration and photosynthesis of *Festuca arundinacea.* The novelty is that we worked with up to three
closed chambers simultaneously in an automated way which allows an exploration of numerous
different plant species and climatic conditions. Moreover, the isotopic analyses are now performed
with an optical spectrometer (Optical-Feedback Cavity-Enhanced Absorption Spectroscopy, i.e. OF-
CEAS technique) in addition to IRMS. This spectrometer allows studying the concentration and the
isotopic composition of $O_2$ in the different chambers in a continuous way.
This manuscript is organized as follows. First, we will present new developments on closed biological
chambers compared to the study of Paul et al. (2023) as well as the multiplexing system integrating
continuous measurements of elemental and isotopic composition of $O_2$. Then, we will present the
results of a biological experiment where photosynthesis and respiration took place. Finally, we will
provide estimate of fractionation factors through two analytical techniques: optical spectrometry and
IRMS.

2.  Material and Methods

2.1.       Growth chamber and closed system

A set of three airtight transparent welded polycarbonate chambers (120 L volume) were adapted from
the chamber described in Paul et al. (2023) and Milcu et al. (2013). The main controlled environmental



parameters inside the closed chambers were temperature, light intensity, $CO_2$ concentration, relative
humidity and differential pressure.
$CO_2$ mixing ratio during light period (dominated by photosynthesis) was regulated with short (30s)
pulses of pure $CO_2$ provided at regular intervals (90s for a sequence with 3 chambers) to each chamber
using a mass flow controller (F200CV, Bronkhorst, The Netherlands) and a Valco selector (EUTF-
SD12MWE, VICI AG International, Switzerland). During the dark period (dominated by plant and soil
respiration), the $CO_2$ is trapped through a 0.5-liter cylinder filled with soda lime and was connected to
a NMS020B KNF micropump.
Unlike the system described in Paul et al. (2023) (Fig.1), relative humidity in each chamber was
controlled using a thermoelectric cooler (100 watt, ET-161-12-08-E Adaptive). The cooled side of the
cooler was in thermal contact with an aluminum rod (1.5 cm diameter) connected to a heat exchanger
acting as condenser inside the chamber. The temperature of the condenser block was monitored with
a thermistor, and the condensed water was directed to the plastic tray containing the plant using an 8
mm plastic tube.
Each chamber was used as a closed gas exchange system, and placed in a separate controlled
environment growth chamber, in the Microcosms experimental platform of the Montpellier European
Ecotron. The temperature of the growth chamber was automatically adjusted in order to keep constant
the temperature at 20°C inside the closed chamber (growth chamber usually set between 20 and 21°C
during dark period and around 18°C during light period because of the greenhouse effect in the
chamber). Air and soil temperature were monitored using 4 NTC probes (CTN 35, Carel). Air relative
humidity and temperature were monitored with a capacitive humidity sensor and a PT100 (PFmini72,
Mitchell Instruments, USA). Air $CO_2$ mixing ratio was monitored using a K30 probes (K30, Senseair).
To find potential leaks in each chamber, helium tests were performed before each experiment.






a)




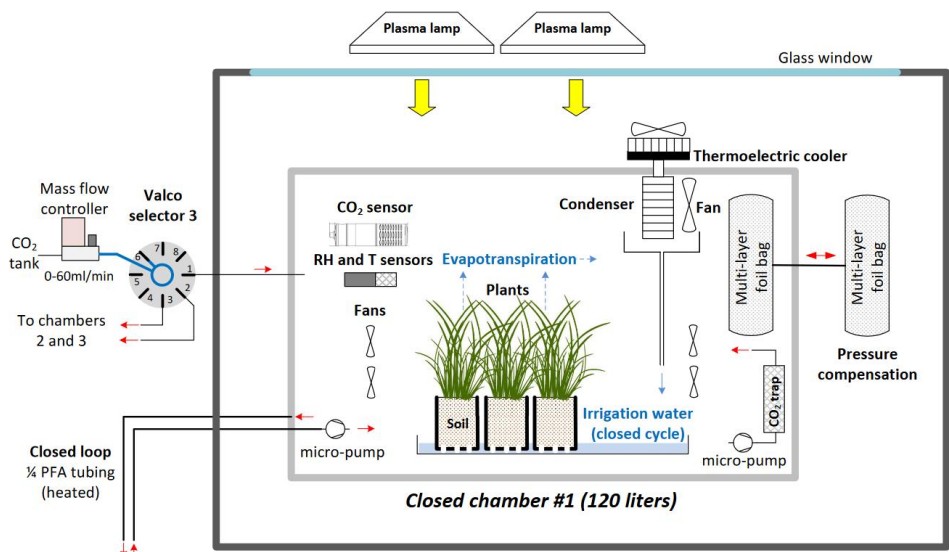


b)

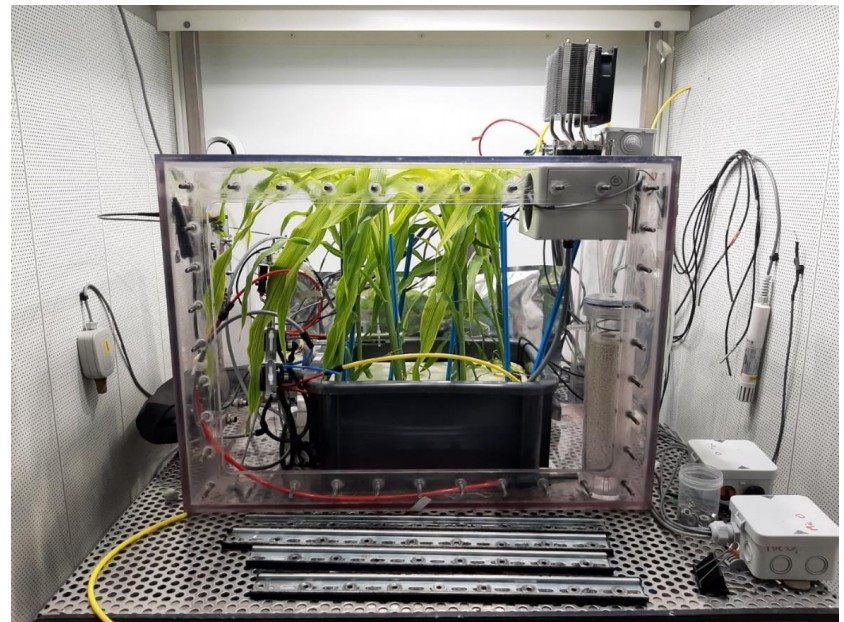




Fig.1. The set-up of the closed chamber system hosting a vegetation-soil atmosphere analogue of the
terrestrial biosphere. (a) Schematic of the closed chamber setup used for the terrestrial biosphere
model. The closed chamber was enclosed in a larger growth chamber. Main environmental parameters
inside the closed chamber were actively controlled and monitored: temperature (T), light intensity,
$CO_2$, relative humidity (RH), pressure differential (ΔP). The water cycle in the closed chamber is shown
in blue. (b) Photograph of the closed chamber used in the experiment with *Zea Mays*.

2.2.       Multiplexing system

With this set-up, we continuously measured the isotopic composition of $O_2$ using an online optical
spectroscopy instrument, hereafter the isotopic analyzer. For each chamber, air circulated through
two external closed loops connected by a tee. The first loop is made of 1/8-inch PFA tubing and used
a Valco selector (12 positions 1/8 inch, EUTF-SD12MWE, VICI AG International, Switzerland) to enable
the air to circulate from one closed chamber through the isotopic analyzer and back to the closed
chamber (Fig.2). The Valco valve selected the origin of the air to be sent to the isotopic analyzer. Five
different origins can be selected (but more can be added): three different closed system chambers and
two reference gases ((1) dried atmospheric air (with a magnesium perchlorate trap), (2) synthetic air
(Alphagaz 2, Air Liquide, France) or dry natural air with 23 % $O_2$ (Natural Air, Air Liquide Espana, Spain)).
Air at the entrance of the isotopic analyzer was dried with a 20 cm long trap (6 mm PFA tube filled with
magnesium perchlorate, renewed daily), and filtered (Millex-FH 0.45 µm/50 mm PTFE hydrophobic
filter, Merck, Germany).





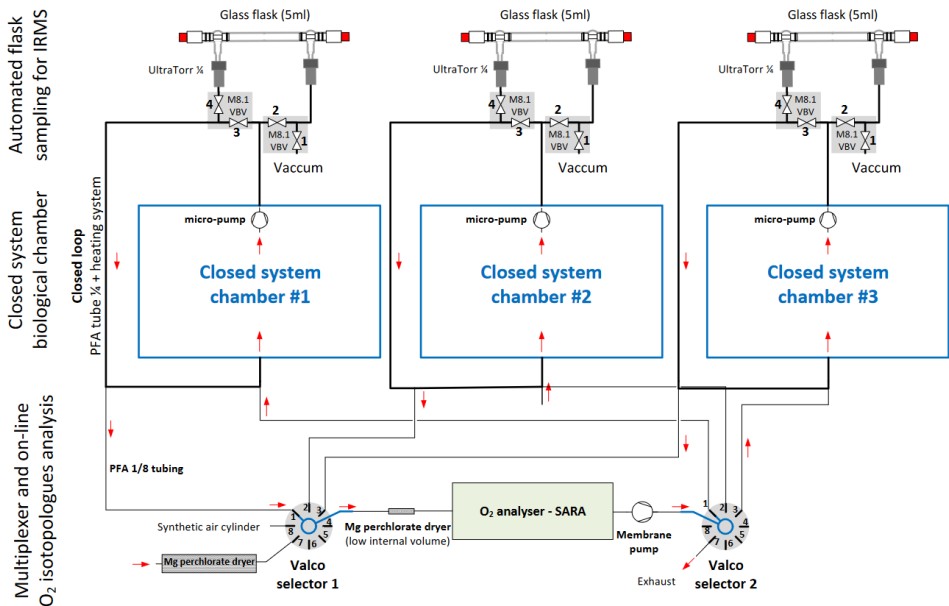


Fig.2. Diagram of the multiplexing system: the enclosed atmosphere of three biological system
chambers circulates through automated flask sampling systems using loops employing ¼" PFA tubing
and micro-pumps. Subsequently, air from these loops is sub-sampled using 1/8-inch PFA tubes and
Valco selectors and analyzed with an isotope analyzer.

Once analyzed, the air stream entered a membrane pump (N811KN.18, KNF, Germany), and
subsequently the common port of a second Valco selector (12 positions 1/8 inch, EUTF-SD12MWE, VICI
AG International, Switzerland). The air was then either redirected to its chamber of origin (closed
circuit) or vented outside of the chamber through an exhaust port for the calibration gases. The
multiplexer composed of two Valco valves ensure three functions : (1) "calibration" : dried ambient air
or synthetic air is provided to the spectrometer, and the outlet is vented to the atmosphere, (2)
"purge": the remaining air still present inside the spectrometer is vented to the atmosphere, until it is
fully replaced by the new stream of air (in order to avoid cross contamination of the air between
chambers, or contamination of a given chamber with the calibration stream), (3) "measurement": the
air sub-sampled from a given chamber is flowing through the spectrometer, and then back to the
chamber. A typical sequence is described in Table 1.



Table 1. Typical measurement sequence with the optical spectrometer. Note that a small amount
(around 5 mL) of air sampled from the chamber is wasted (Valco 2 exhausts to atmosphere) during the
purging phase.

| Phase | Duration (s) | Valco 1 (Port selected) | Valco 2 (Port selected) | Targeted chamber |
|-------|-------------|-------------------------|-------------------------|------------------|
| Calibration | 300 | 7 | 7 | - |
| Purge | 20 | 1 | 7 | 1 |
| Measurement | 280 | 1 | 1 | 1 |
| Calibration | 300 | 7 | 7 | - |
| Purge | 20 | 2 | 7 | 2 |
| Measurement | 280 | 2 | 2 | 2 |
| Calibration | 300 | 7 | 7 | - |
| Purge | 20 | 3 | 7 | 3 |
| Measurement | 280 | 3 | 3 | 3 |


The second loop, used in parallel to the first one described above, is dedicated to the sampling of air
for further analysis by IRMS, as already done in Paul et al. (2023) (Fig.2). Air sampled from each
chamber was circulating continuously into a closed loop (PFA tubing, 1/4-inch, total length between 5
and 10m depending on the chamber location relative to the measurement system) using a micropump
with a flow rate of approximatively 1 L/min (NMS020B, KNF, Germany), through an automated flask
sampling system. All tubes were heated using self-regulating heating cable (15W/m, reference), and
the sampling system was located in a temperature regulated enclosure (25 to 30°C). The sampling
system was made of two three-way pneumatic valves for each chamber (M8.1 VBV, Rotarex)
connected to a glass flask (5mL, as described in Paul et al. (2023)) with two Ultra-Torr fittings (SS-4-UT-
9, Swagelok, USA) and ensured three functions as described in Table 1: (2) "Purge": the flask is isolated
from the closed loop and connected to a vacuum pump (1 to 5 mbar), (2) "Sampling": the air from the
loop is flowing through the sampling flask and back to the loop, (3) "Hold": the flask is isolated from
the closed loop in order to be manually closed and collected. During a typical sequence, each flask was
evacuated ("purge") for 10 minutes, then the "sampling" was activated for at least 30 minutes, and
"hold" was triggered at a time selected by the user using a computer-controlled system (Table 2).





Table 2. Sampling sequence with the flask sampling system.

| State | Valves | | | | Air flow | Duration (min) |
|---|---|---|---|---|---|---|
| | 1 | 2 | 3 | 4 | | |
| Purge | Open | Closed | Open | Closed | Flask bypassed | 10 |
| Sampling | Closed | Open | Closed | Open | Through flask | 30 |
| Hold | Closed | Closed | Open | Closed | Flask bypassed | to be adjusted |


2.2.2. Description of control commands

The control software was developed using open-source Python libraries (PyQt5 for the GUI) and
homemade drivers to interact with the various elements (valves, sensors, regulators, etc.) through
serial connections. It included a user interface displaying the state of relevant components and the
value of the different sensors. The software had three main functions:
- controlling the chamber's $CO_2$ injection rate: the desired $CO_2$ rate target can be set and the automatic
regulation can be turned on or off using the GUI (Graphical User Interface). The $CO_2$ trap state and the
$CO_2$ injection flow rate were also displayed. Real-time plots showed the $CO_2$ in ppm of each chamber,
for quick and easy visual control.
- controlling the flow path of the optical spectrometry analyzer, by sending commands to the upstream
and downstream valves. A sequencer can be used to define how long and in which order the chambers
or calibration gases should be measured by the optical spectrometer.
- controlling the flask sampling. This part controls the pneumatic valves which create the flow path for
purging, filling or holding the content of the flask. The duration of the purge and the absolute
timestamp of the sampling can be set individually for each chamber, for automatic sampling, while
manual operation is still possible.
Furthermore, the control software retrieved concentration data from the optical spectrometer via an
Ethernet connection and merged it with the flow path data into an unified, time-consistent file for
convenient future analysis.





### 2.2.3. Mass spectrometry and optical spectrometry analyses

#### 2.2.3.1. Mass spectrometry analyses technique

In order to be able to compare the evolutions of $\delta^{18}O$ of $O_2$ and $\delta O_2/N_2$ measured by the mass spectrometer and the optical spectrometer during light and dark periods, we collected the air in the chamber via the flask sampling system during one dark period (night 1) and one light period (day 2). We collected 6 flasks for the dark period and 5 flasks for the light period.

The air sampled by the flask system of the second loop was transported to LSCE. The air collected was purified by a semi-automatic separation line (Capron et al., 2010) and analyzed by a Delta V plus dual inlet mass spectrometer (Thermo Electron Corporation). One run consists of 2 series of 16 measurements for each sample and measures the isotopic composition of the air: $\delta^{18}O$ of $O_2$ and $\delta O_2/N_2$ (Extier et al., 2018).

#### 2.2.3.2. Optical spectrometry analyses (OF-CEAS technique)

The description of the OF-CEAS laser optical spectrometer is detailed in Piel et al. (preprint). The spectrometer measured simultaneously $\delta^{18}O$ of $O_2$ and $O_2$ mixing ratio. In our case, because of an experimental problem during the experiment, the instrument was working with a slightly deteriorated precision.

a)






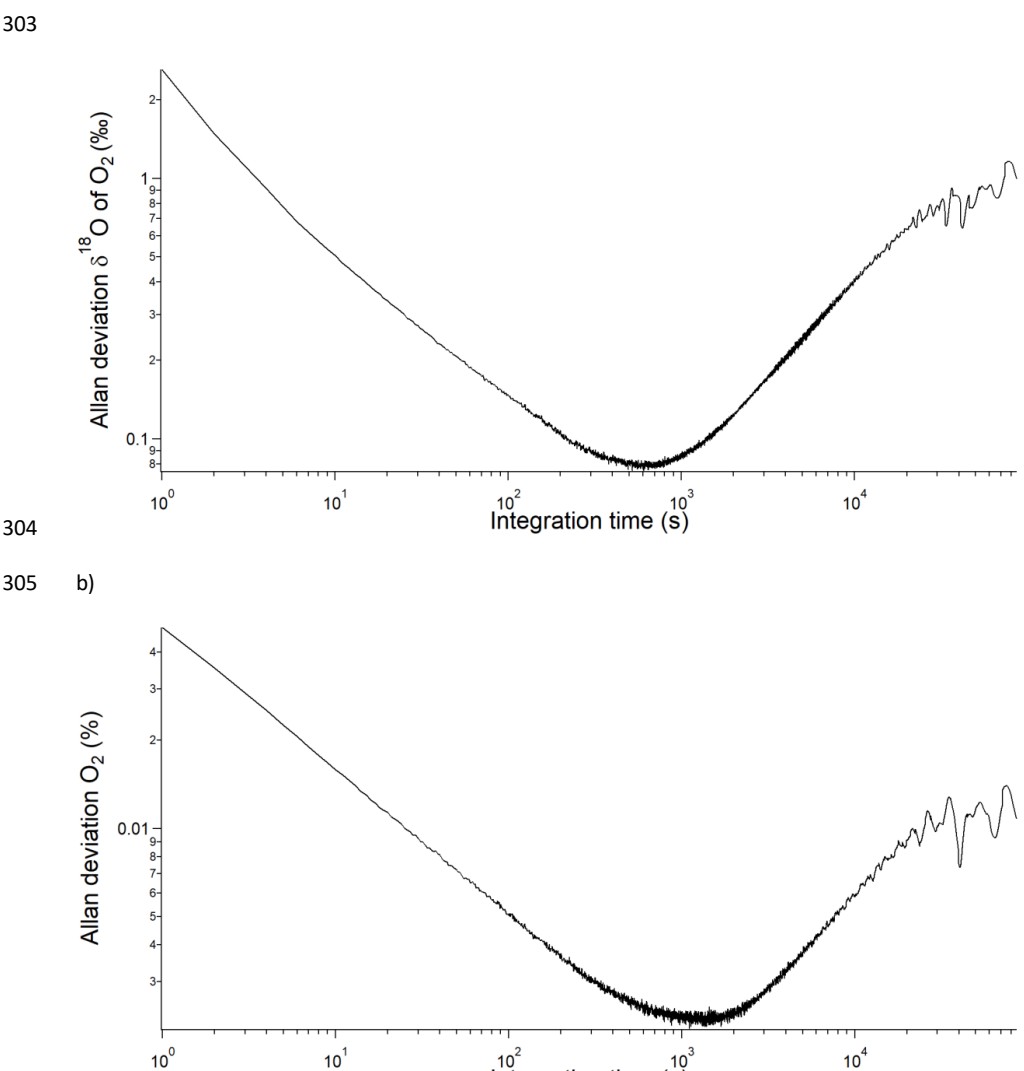


b)

Fig. 3 Allan deviation for (a) $\delta^{18}O$ of $O_2$ and (b) $O_2$ concentration from optical spectrometry during our

studies (i.e. deteriorated mode).

In order to estimate the instrument overall precision versus raw measurement integration time, we

used Allan deviation which is the square root of Allan variance (Werle, 2011). The minimum of the

curve can be interpreted as the best precision the instrument can achieve and the optimum integration

time. In our case (Figure 3), the best precision was 0.08 ‰ and 22 ppm for $\delta^{18}O$ and $O_2$ mixing ratio

respectively, with an optimum integration time of 10 minutes. Furthermore, the $\delta^{18}O$ of $O_2$ level



remains consistently below 0.1 ‰ for a duration of 20 minutes. Based on this trend, we can infer that
calibrating the instrument every 20 minutes would prevent any drift-related issues.

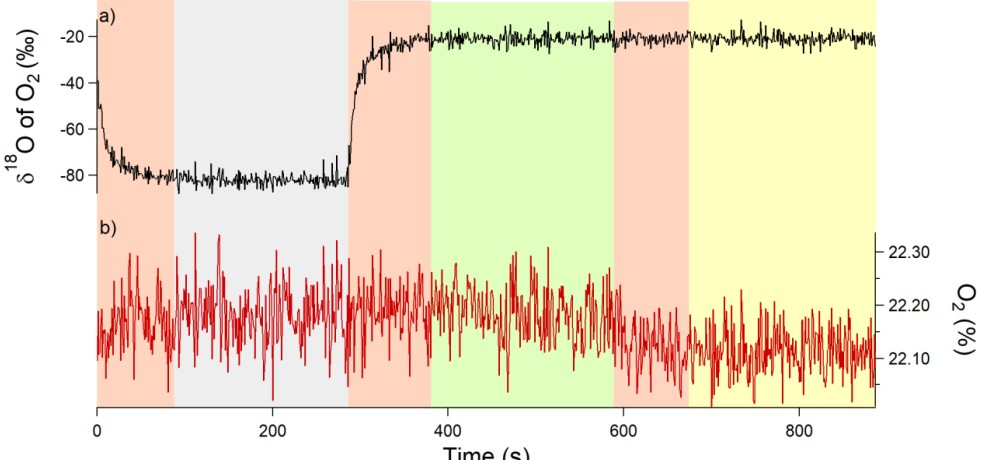


Fig.4 Example of an 18-minute measurement sequence for a closed chamber with two calibrations.
Grey rectangle corresponds to calibration 1, i.e. synthetic air (with a $\delta^{18}O$ of $O_2$ value of - 60 ‰
measured by IRMS and with a $O_2$ concentration of 20.9 %), measured for 6 min. Green rectangle
corresponds to calibration 2, i.e. atmospheric air (with $\delta^{18}O$ of $O_2$ value equal to 0 ‰ and $O_2$
concentration 21 %), measured for 6 min. Yellow rectangle corresponds to the air measurement of the
closed chamber measured for 6 min. All the pink rectangles represent the memory effect of the
analyzer, those measurement points were removed from the processed and analyzed data (i.e. first 2
minutes removed). (a) $\delta^{18}O$ of $O_2$ in black and (b) $O_2$ concentration in red.
For our sequence of measurements, we choose two calibration gases: the atmospheric air which is the
reference gas and a synthetic gas which had an isotopic signature of - 60 ‰ for the $\delta^{18}O$ of $O_2$ and a
concentration of $O_2$ of 23%. The sequence of measurements experiments was then: 6 min of
measurement of synthetic air - 6 min of measurement of atmospheric air - 6 min of measurement of
air in the chamber. This sequence was then applied to each of the 3 chambers and a full sequence
lasted 18 min (Fig.4).
We had a clear memory effect when switching from one gas to another (Fig. 4). As a consequence, we
removed the data of the first 2 minutes before averaging the measurements over the last 4 minutes
(the instrument provided measurements at a frequency of 3 Hz) to get one averaged value. Finally,
there was a dependence of $\delta^{18}O$ of $O_2$ on the concentration of $O_2$ for the spectrometry analyzer and



for this study, the correction for the influence of $O_2$ concentration on $\delta^{18}O$ of $O_2$ is given by:
$\delta^{18}O_{corr} = \delta^{18}O_{measured} - (0.3736 \times [O_2] + 0.0165)$ (details in Piel et al. (preprint)).

2.2.5. Experimental run

We present here the results of one experiment performed on growing maize (*Zea mays* L.) on a typical
compost soil (*Terreau universel*, Botanic, France. Composition: black and blond peat, wood fibre, green
compost and vermicompost manure, organic and organo-mineral fertilizers and micronutrient
fertilizers) in three closed chambers in parallel. The experiment lasted 5 days, with alternating dark
and light periods as follows: day 1 (6 h light) / night 1 (37 h dark) / day 2 (6 h light) / night 2 (56 h dark)
/ day 3 (10 h light). The dark periods were imposed to be longer than the light periods because the
production rate of oxygen during photosynthesis was much stronger than the consumption rate of
oxygen by respiration.

2.2.4. Quantification of fractionation factors associated with respiration and photosynthesis
process

In order to calculate the fractionation factors associated with dark respiration and photosynthesis of
soil and maize, we used the equations 4 and 5 (for details, refer to Paul et al. (2023)).
The isotopic discrimination for dark respiration, $^{18}\varepsilon_{dark\_respi}$, is given by:
$$^{18}\varepsilon_{dark\_respi} = {}^{18}\alpha_{dark\_respi} - 1 = \frac{ln\left(\frac{\delta^{18}O_t + 1}{\delta^{18}O_{t0} + 1}\right)}{ln\left(\frac{n(O_2)_t}{n(O_2)_{t0}}\right)} \tag{4}$$


Where $^{18}\alpha_{dark\_respi}$ is the dark respiration fractionation factor, t0 is the starting time of each dark period
and t is the time of the experiment.
$\frac{n(O_2)_t}{n(O_2)_{t0}}$ is linked to $\delta\left(\frac{O_2}{N_2}\right)$ as:




$$\frac{n(O_2)_t}{n(O_2)_{t0}} = \frac{\frac{\delta\left(\frac{O_2}{N_2}\right)_t}{1000}+1}{\frac{\delta\left(\frac{O_2}{N_2}\right)_{t0}}{1000}+1} \qquad (5)$$




Photosynthesis fractionation factor, $^{18}\alpha_{\text{photosynthesis}}$, is calculated as:

$$^{18}\varepsilon_{photosynthesis} = {}^{18}\alpha_{photosynthesis} - 1$$

$$= \frac{n(O_2)_t / n(O_2)_{t0} \times a^{18}R + {}^{18}R_t \times \left(F_{photosynthesis} - F_{dark\_respi} + {}^{18}\alpha_{dark\_respi} \times F_{dark\_respi}\right)}{{}^{18}R_{lw} \times F_{photosynthesis}} \qquad (6)$$


Where $a^{18}R = \frac{d\,^{18}R}{dt}$ during the light period, $F_{photosynthesis}$ and $F_{dark\_respi}$ are, respectively,
photosynthesis and dark respiration fluxes of oxygen and $lw$ stands for leaf water.
Note that because maize is a C4 plant, we consider that photorespiration and Mehler reaction were
not involved in the $O_2$ consumption by the plant.

## 3. Results

3.1. Comparison between mass-spectrometry and optical-spectrometry analysis










Fig.5 Evolution of the different isotopic ratios of the soil and maize experiment due to dark respiration
and photosynthesis (starting 05/10/22 and ending 10/10/22) in closed chambers over 5 days. Grey
rectangles correspond to dark periods and white rectangles to light periods. (1) corresponds to
chamber 1, (2) chamber 2, (3) chamber 3. (a) $\delta^{18}O$ of $O_2$ variations. (b) $\delta O_2/N_2$ variations. Black points:
optical spectrometer's data (OF-CEAS). Red stars: data obtained by IRMS. Red dashed line: linear
regression of optical spectrometer data for one period (dark or light). Black dashed line: linear
regression of IRMS data for one period (dark or light). Note that the first period of light is not
considered because the system is not stable at that stage.
Figure 5 presents the evolution of the elemental concentration and isotopic composition of dioxygen
in the biological chambers during the experiment described in the previous section. Because of
calibration, averaging and switch from one chamber to another every 18 minutes, the optical
spectrometry analyzer provides only one $\delta^{18}O$ of $O_2$ and $O_2$ concentration value every 54 minutes in
each chamber.
During dark periods, when there was only soil and plant respiration, $\delta^{18}O$ of $O_2$ increased by 1 ‰ and
$\delta O_2/N_2$ decreased by 50-60 ‰ (Fig. 5). During the light period, when both photosynthesis of plant and
respiration in the plant and soil occurred, the $\delta^{18}O$ of $O_2$ decreased by 1 ‰ and $\delta O_2/N_2$ increased by
around 50 ‰ at a rate twice as fast as the decrease of respiration rate observed during nigh periods.
In Figure 5, the optical spectrometer-derived  $\delta^{18}O$ of $O_2$ data displayed a higher degree of scattering
compared to the data obtained through the use of IRMS. Nonetheless, the regression slopes computed
for each period (dark and light period) demonstrate a general comparability, regardless of whether
they are derived from the IRMS or optical spectrometer data (see Table 3). This finding holds significant
importance as the fractionation factors were determined based on the values of these regression
slopes.
Table 3. Average and standard deviation of the isotopic discriminations of maize and the number of
data for all the experiment (with data of the three chambers) on which they were calculated

| Isotopic discriminations of maize | Average (‰) and standard deviation | | Number of data | |
|---|---|---|---|---|
| | IRMS | OF-CEAS | IRMS | OF-CEAS |
| $^{18}\varepsilon_{dark\_respi}$ | - 17.8 ± 0.9 | - 15.9 ± 1.4 | 18 | 249 |
| $^{18}\varepsilon_{photosynthesis}$ | 3.2 ± 2.6 | 6.7 ± 3.8 | 21 | 57 |






From the results displayed on Figure 5, it was possible to calculate the isotopic discrimination found
for dark respiration as $^{18}\varepsilon_{dark\_respi}$ - 17.8 ± 0.9 ‰ and- 15.9 ± 1.4 ‰ for IRMS and optical spectrometer
respectively (Table 4). For photosynthesis, the isotopic discrimination found for $^{18}\varepsilon_{photosynthesis}$, is +
3.2 ± 2.6 ‰ and + 6.7 ± 3.8 ‰, for IRMS and optical spectrometer respectively.
The value of isotopic discrimination, $^{18}\varepsilon_{dark\_respi}$, associated with maize growing on soil agreed with
the literature. Guy et al. (1989) found a value equal to - 17 and - 19 ‰ for $^{18}\varepsilon_{dark\_respi}$ for
Phaeodactylum tricornutum and terrestrial plants. Helman et al. (2005) found a value of $^{18}\varepsilon_{dark\_respi}$
equal to -17.1 ‰ for bacteria from the Lake Kinneret and a value of - 19.4 ‰ for Synechocystis. Paul
et al. (2023), found, for *Festuca arundinacea* a value equal to - 19.1 ± 2.4 ‰. Our value for
$^{18}\varepsilon_{photosynthesis}$ for maize is also close to the value determined by Paul et al. (2023): + 3.7 ± 1.3 ‰ for
*Festuca arundinacea* species. In both cases we observe a positive value. Our value hence confirms the
existence of an apparent isotopic discrimination for terrestrial photosynthesis.

4- Discussion and conclusion

We have presented above a new automated multiplexing system which opens new perspectives for
the study of gas exchange between plant and the atmosphere. In particular, the automation system
has the following advantages.
1-  It provides continuous measurements of the isotopic and elemental composition of dioxygen

in the biological chamber which frees us from the constraint of manual sampling. Moreover, it

provides access to the near-real time evolution of $\delta^{18}O$ of $O_2$ and $O_2$ concentration during the

experiment, in contrast to the delay of the IRMS measurements in classical system. This is

particularly important if an adjustment of the environmental conditions is needed in the

course of the experiment (e.g. duration of dark and light periods).

2-  It permits to run replicate experiments in a very convenient way which opens the way to

systematic studies over a large range of environmental conditions, plant and soil types.

3-  Because our development is associated with open code and classical and relatively low costs

sensors (except for the optical spectrometry analyzer), it is easily adaptable to other biological

experiments. Coupled with other optical spectrometers (e.g. commercial Picarro or Los Gatos

Research (LGR) trace gas instruments optimized for closed systems), this experimental setup



could thus be used to quantify the exchange of trace gases such as $N_2O$ and $CH_4$ (and their
isotopologues) between the plant/soil system and the atmosphere.

When applied to the determination of the fractionation factor associated with dark respiration and
photosynthesis of maize, our results ($^{18}\varepsilon_{dark\_respi}$ of -17 ± 2 ‰ and $^{18}\varepsilon_{photosynthesis}$ of + 6.7 ± 3.8 ‰)
are in general agreement with values found in the literature. Still, the relatively large uncertainty on
the isotopic fractionation factors is due to the fact that our optical spectrometry was not working in
its optimal state and that too much time was devoted to the calibration. It has been found that the
difference in isotopic and elemental compositions of dioxygen for the two calibration gases was stable
over the whole duration of the experiment. Therefore, this suggests that it is enough to measure both
calibration gases only twice every day to check the stability of the linearity and to measure only one
gas of calibration every 15-20 minutes during the day. Future studies should hence make use of an
upgraded instrument and less frequent calibrations.


## Author contributions

AL and CPi designed the project. CPi, JS, SD and CPa carried out experiments at ECOTRON of
Montpellier and FP, CPa, RJ, AD and OJ at LSCE. CPa, CPi and AL analyzed the data from the optical
spectrometer and CPa and AL analysed the data from IRMS. CPa, CPi and AL prepared the manuscript
with contributions from AM.


## Competing interests

The authors declare that they have no conflict of interest.

## Acknowledgements

The research leading to these results has received funding from the European Research Council under
the European Union H2020 Programme (H2020/20192024)/ERC grant agreement no. 817493 (ERC



ICORDA) and ANR HUMI17. The authors acknowledge the scientific and technical support of PANOPLY
(Plateforme ANalytique géOsciences Paris-sacLaY), Paris-Saclay University, France. Our thanks also to
go to AQUA-OXY (CNRS IIT project). This study benefited from the CNRS resources allocated to the
French ECOTRONS Research Infrastructure, from the Occitanie Region and FEDER investments as well
as from the state allocation 'Investissement d'Avenir' AnaEE- France ANR-11-INBS-0001. We would
also like to thank Abdelaziz Faez and Olivier Ravel from ECOTRON of Montpellier for their help and
Emeritus Prof. Phil Ineson from University of York.

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
