# Peer review of "A multiplexing system for quantifying oxygen fractionation factors in"

_EGUsphere, 2024_

## Author Comment (AC1)

Dear Editor,

We are grateful for the invitation to review our manuscript entitled "A multiplexing system for quantifying oxygen fractionation factors in closed chambers". We thank the reviewer because he/she helped us to improve the article. We have made the changes suggested by the reviewer in a version provided below. And, a detailed point-by-point response to the reviewers' comments is provided below.

We hope that you will find this revised manuscript suitable for publication,

On the behalf of all co-authors,

Clémence Paul

**Point-to-point response**

black = reviewer comment / purple = answers / blue = new text / green = unchanged text

**Reply to Referee #1**

Line 91 -92: A sentence explaining how d18O is used to reconstruct oceanic vs terrestrial productivity would be helpful.

Line 105: Replace "despite our system..." with "although our system..."

Line 108-121: Adding a sentence explaining what overall factors are considered when choosing type of species for experimental setup would be helpful

For these three comments on the introduction, in view of referee 2's proposal, we have decided to repeat the introduction in its entirety (without major modifications). The new introduction is given in the second part of the document "reply to referee #2".

Line 236: 'reference' is highlighted, citation missing

The missing reference (15 W m$^{-1}$, Technitrace, France) has been added.

Line 292: "Experimental problem: Please describe what the experimental problem was. Since this is a new method and would potentially be adopted by other labs, it would be useful to know what the problem was and how it was addressed.

The sentence explaining the experimental problem: Liquid water entered the instrument due to condensation in the piping connected to the instrument.

Line 341: Explain why maize was chosen as the preferred option

The dark periods were imposed to be longer than the light periods because the production rate of oxygen during photosynthesis was much stronger than the consumption rate of oxygen by respiration. Maize was chosen as the preferred option, as it is a C4 model plant and enables photosynthetic fluxes

to be clearly differentiated from respiratory fluxes (no photorespiration for C4 plants), so that biological fractionation factors can be calculated easily.

Figures: Adding shaded error bars/ or some sort of way to indicate uncertainty in regression slopes would be visually helpful.

We have calculated the uncertainties in the regression slopes and added them on the graph as uncertainty envelops.

[Figure]

[Figure]

[Figure]

New caption:

Fig.5 Evolution of the different isotopic ratios of the soil and maize experiment due to dark respiration and photosynthesis (starting 05/10/22 and ending 10/10/22) in closed chambers over 5 days. Grey rectangles correspond to dark periods and white rectangles to light periods. (1) corresponds to chamber 1, (2) chamber 2, (3) chamber 3. (a) $\delta^{18}O$ of $O_2$ variations. (b) $\delta O_2/N_2$ variations. Black points: optical spectrometer's data (OF-CEAS). Red stars: data obtained by IRMS. Red dashed line: linear regression of optical spectrometer data for one period (dark or light). Black dashed line: linear regression of IRMS data for one period (dark or light). Pink envelopes represent uncertainty envelops associated with linear regression slopes and intercept of optical spectrometer data for $\delta^{18}O$ of $O_2$. Note that the first period of light is not considered because the system is not stable at that stage.

**Reply to Referee #2**

A scientific article on an innovative but non-exclusive topic deserves a more extensive literature search. This would also make the introduction of the paper more robust.

Thank you for your interesting comment. We add information about Dole effect to have a more detailed and comprehensive introduction. Here's the new 
[revised manuscript text omitted]

Figure 2 has too small, unreadable lettering. I suggest replacing it.

Here a best version of the schema.

[Figure]

In general, I always find it confusing to merge discussions with conclusions. In any case, I suggest adding in the conclusions your group's future perspectives on this very interesting research.

We separate conclusions and perspectives like this:

**4- Discussion**

The value of isotopic discrimination, $^{18}\varepsilon_{dark\_respi}$, associated with maize growing on soil agreed with the literature. Guy et al. (1989) found a value equal to - 17 and - 19 ‰ for $^{18}\varepsilon_{dark\_respi}$ for Phaeodactylum tricornutum and terrestrial plants. Helman et al. (2005) found a value of $^{18}\varepsilon_{dark\_respi}$ equal to -17.1 ‰ for bacteria from the Lake Kinneret and a value of - 19.4 ‰ for *Synechocystis*. Paul et al. (2023), found, for *Festuca arundinacea* a value equal to - 19.1 ± 2.4 ‰.

Our value of $^{18}\varepsilon_{photosynthesis}$ for maize is also close to the value determined by Paul et al. (2023): + 3.7 ± 1.3 ‰ for *Festuca arundinacea* species. In both cases we observe a positive value which contradicts the value classically used of 0 ‰ from Guy et al. (1993). Our value hence confirms the existence of an apparent isotopic discrimination for terrestrial photosynthesis. This leads to an increase of the $\delta^{18}O$ of $O_2$ value associated with terrestrial biosphere compared to the latest study of Luz and

Barkan (2011). As a consequence, it is still an open question to know of $\delta^{18}O_{atm}$ or Dole effect variations should be interpreted solely as a change in the low latitude atmospheric water cycle or if the relative change in the marine vs terrestrial biological productivity also plays a role. Future studies should hence use a set-up similar to ours to systematically study the $O_2$ fractionation coefficients associated with biological processes.

**5- Conclusions and perspectives**

We have developed and presented a new automated multiplexing system that facilitates the study of gas exchange between plants and the atmosphere. This system offers several key advantages. First, it allows continuous measurements of the isotopic and elemental composition of dioxygen in the biological chamber, removing the need for manual sampling. Second, it provides near-real-time monitoring of $\delta^{18}O$ of $O_2$ and $O_2$ concentration during experiments, enabling adjustments to environmental conditions, such as dark and light cycles, in real time. Finally, it supports the convenient replication of experiments, enabling systematic studies across a wide range of environmental conditions, plant species, and soil types.

In the application of this system to maize, the fractionation factors for dark respiration ($^{18}\varepsilon_{dark\_respi}$ : - 17 ± 2 ‰) and photosynthesis ($^{18}\varepsilon_{photosynthesis}$ : + 6.7 ± 3.3 ‰) are consistent with literature values, though the relatively large uncertainties highlight some current limitations, including suboptimal performance of the optical spectrometry and excessive calibration time. Stability tests of the calibration gases indicated that less frequent calibrations (e.g., measuring both gases twice daily and one calibration gas every 15 – 20 minutes) would be sufficient to ensure accuracy.

Our automated system has significant potential for broader applications. First, its open-code design and use of relatively low-cost sensors (excluding the optical spectrometry analyzer) make it easily adaptable to other biological experiments. Second, coupling this system with other optical spectrometers, such as Picarro or Los Gatos Research (LGR) trace gas instruments, could enable the quantification of trace gas exchanges, including $N_2O$ and $CH_4$ (and their isotopologues), between the plant/soil system and the atmosphere.

Future studies should focus on upgrading the instrumentation to enhance performance and reduce uncertainties in isotopic fractionation measurements. Additionally, optimizing calibration frequency will improve experimental efficiency and reliability. This system paves the way for more comprehensive and systematic investigations into gas exchange processes under diverse conditions.